# *APE1/Ref-1* Inhibits Adipogenic Transcription Factors during Adipocyte Differentiation in 3T3-L1 Cells

**DOI:** 10.3390/ijms24043251

**Published:** 2023-02-07

**Authors:** Eun-Ok Lee, Hee-Kyoung Joo, Yu-Ran Lee, Sungmin Kim, Kwon-Ho Lee, Sang-Do Lee, Byeong-Hwa Jeon

**Affiliations:** 1Research Institute of Medical Sciences, College of Medicine, Chungnam National University, 266 Munhwa-ro, Daejeon 35015, Jung-gu, Republic of Korea; 2Department of Medical Science, College of Medicine, Chungnam National University, 266 Munhwa-ro, Daejeon 35015, Jung-gu, Republic of Korea; 3Department of Physical Therapy, Joongbu University, 201 Daehak-ro, Geumsan-gun 32713, Chungcheongnam-do, Republic of Korea

**Keywords:** *APE1/Ref-1*, adipocyte differentiation, adipogenic transcription factor, *C/EBP-α*, *PPAR-γ*, 3T3-L1 cells

## Abstract

Apurinic/apyrimidinic endonuclease 1/redox factor-1 (*APE1/Ref-1*) is a multifunctional protein involved in DNA repair and redox regulation. The redox activity of *APE1/Ref-1* is involved in inflammatory responses and regulation of DNA binding of transcription factors related to cell survival pathways. However, the effect of *APE1/Ref-1* on adipogenic transcription factor regulation remains unknown. In this study, we investigated the effect of *APE1/Ref-1* on the regulation of adipocyte differentiation in 3T3-L1 cells. During adipocyte differentiation, *APE1/Ref-1* expression significantly decreased with the increased expression of adipogenic transcription factors such as CCAAT/enhancer binding protein (*C/EBP*)-*α* and peroxisome proliferator-activated receptor (*PPAR*)-*γ*, and the adipocyte differentiation marker adipocyte protein 2 (*aP2*) in a time-dependent manner. However, *APE1/Ref-1* overexpression inhibited *C/EBP-α*, *PPAR-γ,* and *aP2* expression, which was upregulated during adipocyte differentiation. In contrast, silencing *APE1/Ref-1* or redox inhibition of *APE1/Ref-1* using E3330 increased the mRNA and protein levels of *C/EBP-α*, *PPAR-γ*, and *aP2* during adipocyte differentiation. These results suggest that *APE1/Ref-1* inhibits adipocyte differentiation by regulating adipogenic transcription factors, suggesting that *APE1/Ref-1* is a potential therapeutic target for regulating adipocyte differentiation.

## 1. Introduction

Adipocytes play important roles in energy metabolism and lipid storage [1]. The growth of adipose tissue involves an increase in adipocyte size and differentiation of adipocytes from preadipocyte precursor cells [2]. Under conditions of excess body fat and obesity, the body accumulates excessive amounts of triglycerides in adipocytes, which increases the triglyceride content of the liver, muscle, adipose tissue, and plasma, leading to pathological dysfunctions, such as metabolic syndrome, coronary heart disease, hypertension, type 2 diabetes, cancer, and osteoarthritis [3,4].

Apurinic/apyrimidinic endonuclease 1/redox factor-1 (*APE1/Ref-1*) is a multifunctional protein involved in DNA repair and the redox regulation of gene expression. The C-terminal domain of *APE1/Ref-1* processes the apurinic/apyrimidinic endonuclease activity of the base excision repair pathway, which is involved in cellular responses to oxidative stress as well as maintenance of genome stability and transcriptional activity [5,6,7]. The N-terminal domain of *APE1/Ref-1* possesses redox activity, which regulates transcription factors through proton exchange with cysteine residues in various cells and regulates the expression of genes that directly affect several cellular processes, including the inflammatory response [8,9]. *APE1/Ref-1* haploinsufficiency mice showed genomic instability, contributing to increased apoptotic response or atypical cell differentiation [10,11].

Mature adipocytes differentiate from preadipocytes upon exposure to adipogenic hormones, including insulin, and through activation by adipogenic transcription factors [12]. Therefore, studying the mechanisms of gene regulation and transcription factor activity involved in preadipocyte differentiation is important for the prevention and treatment of obesity and obesity-related diseases. Transcription factors involved in the differentiation of preadipocytes include CCAAT/enhancer binding protein (*C/EBP*)-β, *C/EBP-*δ, *C/EBP-α*, peroxisome proliferator-activated receptor (*PPAR*)-*γ*, and sterol regulatory element-binding binding protein 1 (*SREBP-1*). Among these, *C/EBP-α* and *PPAR-γ* are strongly expressed in response to adipocyte differentiation signals before triglyceride production [13,14,15]. Furthermore, the degree of adipocyte differentiation is closely related to the level of adipocyte protein 2 (*aP2*) [16].

*APE1/Ref-1* is involved in regulating the differentiation of hematopoietic cells, dental progenitor cells, and vascular smooth muscle cells [6,7,17]. Hemangioblast development from embryonic stem cells is reduced when *APE1/Ref-1* is knocked down; furthermore, inhibition of *APE1/Ref-1* redox activity using E3330 inhibits hemangioblast development [17]. Inhibition of *APE1/Ref-1* redox regulation can also promote the osteo/odontogenic differentiation capacity of dental papilla cells [7]. Recently, *APE1/Ref-1* has been reported to inhibit inorganic phosphate-induced osteoblastic differentiation by modulating osteogenic transcription factors, such as runt-related transcription factor 2 and pituitary-specific positive transcription factor 1 [6]. Failure of DNA repair by *APE1/Ref-1* leads to genomic instability [18], and it is related to lipodystrophies in subcutaneous white adipose tissues [19]. The biological effect of *APE1/Ref-1* as a redox coactivator for early growth response protein-1 (*Egr-1*) transcriptional activity is related with downregulation of Egr-1 target gens such as *C/EBP-β* in mammalian cells [20]. However, it is unknown whether *APE1/Ref-1* can regulate adipogenesis or adipogenic transcriptional factors.

*APE1/Ref-1*, which regulates the activity of various transcriptional regulators [17], is expected to have the potential to regulate the expression of adipogenic transcription factors. Here, we evaluated the role of *APE1/Ref-1* in adipocyte differentiation and its underlying mechanisms in 3T3-L1 cells.

## 2. Results

### 2.1. APE1/Ref-1 Expression Decreased during Adipocyte Differentiation

To determine the changes in *APE1/Ref-1* expression during the adipogenic differentiation of preadipocytes, we investigated *APE1/Ref-1* expression on days 0, 2, 4, 6, and 8 of 3T3-L1 cell differentiation (Figure 1A). Differentiation of 3T3-L1 cells to mature adipocytes was induced using differentiation medium MDI for 8 days and was confirmed by staining with Oil Red O (Figure 1B). The mRNA expression of *APE1/Ref-1* decreased by 58% from days 4 to 8 in a time-dependent manner during differentiation (Figure 1C), whereas the mRNA levels of adipocyte differentiation markers (*C/EBP-α*, *PPAR-γ*, and *aP2*) in MDI-treated cells were significantly elevated compared to those in the untreated cells (Figure 1D–F). Next, we investigated the protein expression of APE1/Ref-1, C/EBP-α and aP2 using Western blotting. As shown in Figure 1G, the protein level of APE1/Ref-1 showed the same decreasing tendency as the mRNA level of *APE1/Ref-1* during adipocyte differentiation (Figure 1H); however, the protein levels of C/EBP-α and aP2 increased from days 4 to 8 after differentiation, which was inversely proportional to APE1/Ref-1 protein expression (Figure 1I,J). These results indicate that *APE1/Ref-1* expression was reduced during adipocyte differentiation in 3T3-L1 cells. Therefore, it is suggested that downregulation of *APE1/Ref-1* may play an important role in adipocyte differentiation.

### 2.2. APE1/Ref-1 Overexpression Decreased Adipocyte Differentiation

To evaluate the effect of *APE1/Ref-1* on MDI-induced adipocyte differentiation, 3T3-L1 cells were transfected with adenovirus expressing *APE1/Ref-1*. The cells were pretreated with adenovirus expressing β-gal or adenovirus expressing *APE1/Ref-1* for 24 h, and adipocyte differentiation was induced using MDI for the subsequent 8 days. In adenoviral *APE1/Ref-1*-transfected cells, *APE1/Ref-1* was overexpressed at both mRNA and protein levels, as detected using qRT-PCR and Western blotting, respectively (Figure 2A,B). The mRNA levels of *C/EBP-α*, *PPAR-γ*, and *aP2* increased in MDI-induced 3T3-L1 cells, but overexpression of *APE1/Ref-1* significantly suppressed the increased *C/EBP-α*, *PPAR-γ*, and *aP2* mRNA levels (Figure 2C–E). We then investigated the protein expression in AdAPE1/Ref-1-transfected 3T3-L1 cells using Western blotting (Figure 2G). The protein expression of both C/EBP-α and aP2 was significantly enhanced in the MDI-treated group (Figure 2F); however, overexpression of *APE1/Ref-1* reduced the expression of C/EBP-α and aP2 in MDI-induced 3T3-L1 cells, indicating that *APE1/Ref-1* exerted an inhibitory effect on adipocyte differentiation (Figure 2G,H). These results indicate that *APE1/Ref-1* overexpression inhibits adipocyte differentiation by suppressing the expression of adipocyte transcription factors such as *C/EBP-α* or *PPAR-γ*.

### 2.3. Silencing APE1/Ref-1 Promotes Adipocyte Differentiation

Having established that overexpression of *APE1/Ref-1* suppressed adipocyte differentiation, the effect of gene silencing of *APE1/Ref-1* during MDI-induced adipocyte differentiation was evaluated in 3T3-L1 cells with transfection of *APE1/Ref-1* siRNA. *APE1/Ref-1* siRNA-transfected cells were exposed to 50 nM for 24 h, and the control group cells were treated with negative control siRNA at the same concentration. Adipocyte differentiation was induced using MDI for 8 days. As shown in Figure 3A,B, the cells transfected with siAPE1/Ref-1 showed significantly reduced mRNA and protein expression of *APE1/Ref-1*. The expression of *APE1/Ref-1* was confirmed to be the lowest in the group induced with MDI following transfection of siAPE1/Ref-1 into 3T3-L1 cells. As shown in Figure 3C–E, silencing *APE1/Ref-1* significantly increased the mRNA levels of *C/EBP-α* and *PPAR-γ*. As shown in Figure 3G,H *APE1/Ref-1* silencing also induced upregulation of C/EBP-α and aP2 at the protein level. These results suggest that *APE1/Ref-1* silencing promotes adipocyte differentiation of 3T3-L1 cells.

### 2.4. Redox Inhibition of APE1/Ref-1 Augments Adipocyte Differentiation

The redox function of *APE1/Ref-1* is involved in the regulation of several kinds of transcriptional factors [20]. Finally, the effect of E3330, a redox inhibitor of APE1/Ref-1 [21], on adipocyte differentiation was evaluated in 3T3-L1 cells.

The cells were incubated with 50 µM E3330 for 24 h before induction of differentiation using the MDI cocktail. As shown in Figure 4A,B, treatment with E3330 did not affect mRNA or protein levels of *APE1/Ref-1*. Redox inhibition of APE1/Ref-1 using E3330 increased the mRNA levels of *C/EBP-α*, *PPAR-γ*, and *aP2* (Figure 4C–E) during adipocyte differentiation in MDI. C/EBP-α and aP2 protein levels were also significantly upregulated by the redox inhibitor of *APE1/Ref-1* (Figure 4G,H). E3330 itself did not affect APE1/Ref-1 expression (Figure 4F), and redox inhibition of APE1/Ref-1 by E3330 increased the mRNA and protein levels of adipogenic transcription factors and *aP2* expression. These results indicate that the redox function of *APE1/Ref-1* plays a key inhibitory role in adipocyte differentiation by suppressing the expression of adipogenic transcription factors in 3T3-L1 cells.

## 3. Discussion

In the present study, we demonstrated that *APE1/Ref-1* inhibits adipocyte differentiation by suppressing the expression of transcription factors such as *C/EBP-α*, *PPAR-γ*, and *aP2* in 3T3-L1 cells. Our results showed that overexpression of *APE1/Ref-1* reduced *aP2* production via inhibition of transcription factor expression. Conversely, gene silencing of *APE1/Ref-1* increased adipocyte differentiation by increasing the expression of adipogenic transcription factors. Inhibition of the redox function of APE1/Ref-1 by E3330 increased adipocyte differentiation by upregulating the expression of *C/EBP-α*, *PPAR-γ*, and *aP2*. Therefore, the redox function of *APE1/Ref-1* appears to play an important role in adipocyte differentiation in 3T3-L1 cells.

Differentiated adipocytes are triacylglycerol storage depots that store energy and mobilize it when needed [22]. However, as excessive fat cells lead to conditions such as obesity, insulin resistance, non-insulin-dependent diabetes, and cardiovascular disease [23], maintaining their correct amounts is necessary through proper regulation of adipocyte differentiation. On the basis of the results of this study, *APE1/Ref-1* might play a key inhibitory role in obesity-related diseases. However, the biological roles of *APE1/Ref-1* in obesity animal models such as metabolic syndrome have not been reported yet, and it is thought that those studies should be conducted in the future.

There is some evidence that *APE1/Ref-1* inhibits differentiation by acting as a transcription cofactor [6,7,24]. Adipogenesis is a multistep process that is regulated by several transcriptional factors. Among the transcriptional factors, PPAR-γ and C/EBP-α are key factors in the regulation of adipogenesis [25]. Reactive oxygen species (ROS) play an important role in adipocyte differentiation; furthermore, adipogenesis is suppressed by antioxidants or ROS scavengers [26]. Inhibition of ROS production may be one of the mechanisms by which *APE1/Ref-1* inhibits adipocyte differentiation. Zhou et al. reported that knocking out of *Stard3* resulted in the inhibition of adipogenesis with decreased mitochondrial ROS production [27]. Lee et al. also demonstrated that pycnogenol, a group of flavonoids with antioxidants effect, inhibits lipid accumulation in 3T3-L1 cells via suppression of oxidative stress. Sekiya et al. showed that oxidative stress led to lipid accumulation by activating sterol regulatory element-binding protein 1c, one of the transcription factors involved in fatty acid biosynthesis in HepG2 cells [28]. Overexpression of *APE1/Ref-1* inhibits intracellular ROS production via inhibition of nicotinamide adenine dinucleotide phosphate (NADPH) oxidase activity or mitochondrial ROS [5,29,30]. Redox inhibition by *APE1/Ref-1* with cysteine mutants has been reported to increase intracellular ROS levels [6,31], consistent with the results of adipogenesis promotion. Several studies have confirmed that the redox function of *APE1/Ref-1* is important for cell differentiation. A previous study reported that inhibiting the redox function of *APE1/Ref-1* during the differentiation of neurogenic embryonic carcinoma cells increased the differentiation of stem cells to neurogenic phenotypes [32]. Furthermore, Chen et al. demonstrated that inhibition of *APE1/Ref-1* redox regulation promoted the osteo/odontogenic differentiation capacity of dental papilla cells via the canonical *Wnt* signaling pathway [7]. The redox function of *APE1/Ref-1* prevents vascular smooth muscle calcification by inhibiting oxidative stress and osteoblastic differentiation, thereby preventing altered osteoblastic phenotypes in vascular smooth muscle cells [6].

Generally known as 3T3-L1 cells with low transfection efficiency, a previous report showed that successful introduction of the gene in 3T3-L1 with adenoviruses requires a high titer of adenovirus [33]. In the present study, we also used high multiplicities of infection of adenovirus as shown in Material and Methods, because of low expression of the coxsackievirus and adenovirus receptors of 3T3-L1 cells [34]. Researchers who wish to utilize adipocyte culture or primary adipocyte culture need to develop a transgenic mouse line expressing adipose tissue-specific coxsackievirus and adenovirus receptor proteins.

In the present study, through the analysis of adipogenic factors, we showed that the redox function of *APE1/Ref-1* contributed to the reduced expression of *C/EBP-α* and *PPAR-γ.* The redox function of *APE1/Ref-1* has been studied as an important aspect in protecting cells from oxidative stress [11,35]. Specific inhibition of redox activity of APE1/Ref-1 by E3330, a redox inhibitor of APE1/Ref-1, enhanced ROS production [36,37]. In this study, we demonstrated that the anti-adipogenic effect of *APE1/Ref-1* could be mediated by the inhibition of expression of adipogenic transcriptional factors such as *C/EBP-α* and *PPAR-γ*. Furthermore, the redox function of *APE1/Ref-1* contributes to the negative regulation of adipogenesis through inhibition of expression of adipogenic transcriptional factors. Uncovering the anti-adipogenic role of *APE1/Ref-1* would contribute to the drug development for metabolic diseases. Drugs such as insulin, which are related with *APE1/Ref-1* reduction or the use of E3330, inhibiting *APE1/Ref-1* redox function, would promote adipocyte differentiation or obesity, which should be considered in the new therapeutic target.

This study has some limitations. Only the effects of *APE1/Ref-1* and redox inhibition on the induction of preadipocyte differentiation were analyzed in this study. However, investigating the specific role of *APE1/Ref-1* in regulating the differentiation of white and brown adipocytes is necessary. Due to it high metabolic activity, brown adipose tissue has become a promising therapeutic target for obesity or associated metabolic disorders. Recent advances in brown adipose tissue suggest an important role in controlling energy metabolism, which is a promising therapeutic approach to obesity in animal models or humans. Therefore, it is necessary to investigate the role of *APE1/Ref-1* in the regulation of uncoupling protein-1, a unique mitochondrial membrane protein for thermogenesis, and adipogenic transcriptional factors in brown adipose cells. As identifying the intercellular interactions in cultured pre-adipocytes is difficult, an integrated investigation using tissue-specific conditional *APE1/Ref-1*-knockout animals is required to explore this aspect. Especially, the development of adipose tissue-specific *APE1/Ref-1* knockout animals will contribute to identifying the biological role of *APE1/Ref-1* in adipose tissue.

In conclusion, our study demonstrated that endogenous *APE1/Ref-1* is downregulated during adipocyte differentiation, whereas overexpression of *APE1/Ref-1* inhibits adipocyte differentiation; in contrast, silencing *APE1/Ref-1* and redox inhibition using E3330 promotes adipocyte differentiation through the modulation of expression of adipogenic transcriptional factors. Overall, these results suggest that the redox function of *APE1/Ref-1* exerts an anti-adipogenic function by inhibiting adipocyte differentiation.

## 4. Materials and Methods

### 4.1. Cells and Reagents

3T3-L1 cells were purchased from ATCC (Manassas, VA, USA). Dexamethasone, isobutyl-methyl-xanthine, insulin, (E)-3-(2-[5,6-dimethoxy-3-methyl-1,4-benzoquinonyl]) -2-nonyl propenoic acid (E3330), and Oil Red O were purchased from Sigma-Aldrich (St. Louis, MO, USA). Dulbecco’s modified Eagle medium (DMEM) and bovine calf serum (BCS) were obtained from Welgen (Gyeongsan, Gyeongsangbukdo, Republic of Korea). Fetal bovine serum was purchased from Gibco (Grand Island, NY, USA). Lipofectamine RNAiMAX was purchased from Thermo Fisher Scientific (San Jose, CA, USA). Small-interfering RNA of *APE1/Ref-1* (siAPE1/Ref-1), small-interfering RNA of negative control (siNC), and oligo dT primers were purchased from Bioneer (Daejeon, Republic of Korea).

### 4.2. Culture and Adipocyte Differentiation of 3T3-L1 Cells

3T3-L1 cells were cultured in growth medium (DMEM with 10% BCS and 1% penicillin–streptomycin) and incubated under humid conditions at 37 °C with 5% CO_2_ [38]. The cells were grown to confluence in a six-well plate (2 × 10^5^ cells/well). Two days post-confluence (day 0), the medium was changed to DMEM supplemented with 10% fetal bovine serum (FBS) and a differentiation medium (MDI) including dexamethasone (1 µM), isobutyl-methyl-xanthine (500 µM), and insulin (1 µg/mL), and incubated for 2 days. On day 2, the medium was replaced with DMEM supplemented with 10% FBS and 1 μg/mL insulin, and the cells were cultured for 48 h. From day 4, the medium was replaced with DMEM supplemented with 10% FBS every 48 h for up to 8 days.

### 4.3. Oil Red O Staining

To evaluate the accumulation of lipid droplets in differentiated adipocytes, Oil Red O staining was performed on day 8 of culture after medium replacement with differentiation medium [39,40]. 3T3-L1 adipocytes were washed with phosphate-buffered saline (PBS), fixed in 4% formaldehyde, and then washed twice with water. An Oil Red O stock solution was dissolved at 250 mg/50 mL 2-propanol. The working solution was obtained by diluting the stock solution 2:3 with distilled water and filtering thrice through a 0.45 µm filter paper, yielding 0.2% Oil Red O in 40% 2-propanol. The working solution was freshly prepared for each experiment and was filtered immediately before use. To stain the lipids, the cells were incubated with the working solution for 1 h at room temperature; the plates were then washed with PBS and dried [41]. The stained lipid droplets were imaged using a microscope (Leica DM4000B) and DFC420 camera (Leica Microsystems SAS, Wetzlar, Germany). Three independent experiments were performed.

### 4.4. Transfection of Small-Interfering RNA

To evaluate the effects of silencing *APE1/Ref-1* on adipocyte differentiation, 3T3-L1 cells (2 × 10^5^ cells/well) were transfected with 50 nM siRNA targeting mouse *APE1/Ref-1* (5′-GGA GGC AGC GCA GUA AAC A-3′) or 50 nM negative control siRNA using Lipofectamine RNAiMAX, according to the manufacturer’s instructions. One day after transfection, the medium was replaced with differentiation medium to induce adipocyte differentiation.

### 4.5. Transfection of Adenoviruses

3T3-L1 cells were transfected with adenovirus expressing full-length *APE1/Ref-1* (AdAPE1/Ref-1) in a growth medium at 37 °C [6,42]. One day after infecting the cells with adenovirus at multiplicity of infection (MOI) of 10,000 for 24 h, the cells were induced to differentiate to adipocytes using MDI. Total adenoviral transfection was balanced using the same concentration of adenoviruses expressing *β-galactosidase* (Adβ-gal) as the control.

### 4.6. Western Blot Analysis

After adipocyte differentiation, the cells were washed with PBS and harvested in a reporter lysis buffer. The cells were then lysed with RIPA buffer containing a protease inhibitor. The lysates were cleared via centrifugation at 12,000× *g* for 20 min; then, 20 µg of total protein was separated on a 5% or 12% sodium dodecyl sulfate-polyacrylamide electrophoresis gel and transferred onto a polyvinylidene fluoride membrane. After blocking with 5% skim milk for 1 h at room temperature, the blots were incubated overnight at 4 °C with a 1:1000 dilution of the appropriate primary antibody, followed by incubation with the appropriate horseradish peroxidase-conjugated secondary antibody for 1 h at room temperature [43]. Protein bands were detected using an enhanced chemiluminescence detection kit (Amersham Pharmacia Biotech, Piscataway, NJ, USA). All experiments were performed three times.

### 4.7. Quantitative Real-Time Reverse Transcription-Polymerase Chain Reaction (qRT-PCR)

Total RNA from 3T3-L1 cells was isolated using a RNeasy Mini Kit (Qiagen, Valencia, CA, USA). Complementary DNA was synthesized using a reverse transcription PCR kit (iNtRON Biotechnology, Gyeonggido, Republic of Korea). The mRNA levels of *APE1/Ref-1*, *C/EBP-α*, *PPAR-γ*, and *aP2* were evaluated using qRT-PCR with the SYBR Green PCR Master Mix (Promega, Madison, WI, USA). qRT-PCR was performed according to the manufacturer’s protocol using QuantStudio 5 Real-Time PCR (Thermo Fisher Scientific, Waltham, MA, USA). The relative level of target mRNA expression was quantified using the ΔCt method, and glyceraldehyde 3-phosphate dehydrogenase (*GAPDH)* was used as an internal control. All experiments were performed in duplicate and repeated three to four times. Three independent experiments were performed. The primer sequences for qRT-PCR were as follows: 5′-CCT CAC CCA GTG GCA AAT CTG-3′ and 5′-TCC ACA TTC CAG GAG CAT ATC T-3′ for *APE1/Ref-1*; 5′-GGA AGA CCA CTC GCA TTC CTT-3′ and 5′-GTA ATC AGC AAC CAT TGG GTC A-3′ for *PPAR-γ*; 5′-AAG GTG AAG AGC ATC ATA ACC CTA-3′ and 5′-TCA CGC CTT TCA TAA CAC ATT CC-3′ for *aP2*; 5′-CAA GAA CAG CAA CGA GTA CCG-3′ and 5′-GTC ACT GGT CAA CTC CGC AC-3′ for *C/EBP-α*; and 5′-AGG TCG GTG TGA ACG GAT TTG-3′ and 5′-TGT AGA CCA TGT AGT TGA GGT CA-3′ for *GAPDH*.

### 4.8. Statistical Analysis

Values are expressed as mean ± standard error of the mean (SEM). Statistical significance of the differences in measured variables between the basal and treated groups was determined using the one-way ANOVA followed by Dunnett’s or Bonferroni’s multiple comparison tests. Statistical significance was set at *p* < 0.05. All statistical analyses were performed using GraphPad Prism 9.0 software (GraphPad Software, Inc., San Diego, CA, USA).

## Figures and Tables

**Figure 1 ijms-24-03251-f001:**
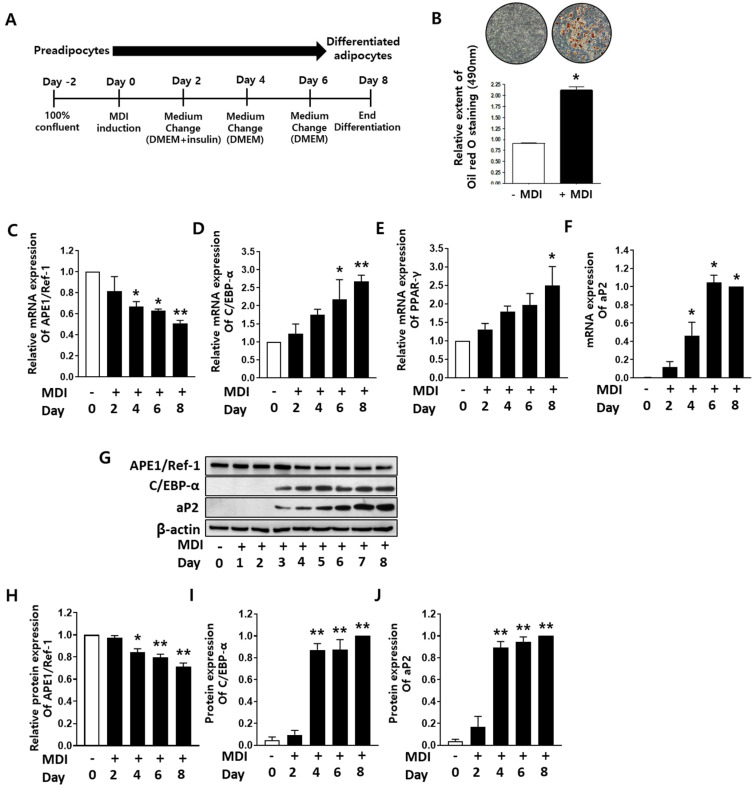
Changes in the expression of adipocyte differentiation markers and apurinic/apyrimidinic endonuclase1/redox factor-1 (*APE1/Ref-1*) during adipocyte differentiation in 3T3-L1 cells. Adipocyte differentiation of 3T3-L1 cells was induced by incubation in MDI during the indicated days as described in Material and Methods. (**A**) Schematic protocol for preadipocyte differentiation of 3T3-L1 adipocytes. (**B**) Oil red O staining after adipocyte differentiation induction with MDI. In the microscopic image (×100 magnification), triglyceride lipid droplets stained with Oil Red O appear red along with a quantitative graph of lipid content obtained using spectrophotometry based on the absorbance at 490 nm. (**C**) A decrease in *APE1/Ref-1* mRNA expression during adipocyte differentiation for the indicated times (0–8 days). (**D**–**F**) Changes in the mRNA expression of adipocyte differentiation markers *C/EBP-α*, *PPAR-γ*, and *aP2*. (**G**) Representative Western blotting results of APE1/Ref-1, C/EBP-α, and aP2 proteins during adipocyte differentiation. (**H**) APE1/Ref-1 protein expression decreased during adipocyte differentiation, and (**I**,**J**) C/EBP-α and aP2 protein expression increased during adipocyte differentiation. The graph shows the average band intensity normalized to β-actin expression. mRNA expression of *aP2* and protein expression of C/EBP-α and aP2 were determined by comparing with the relative maximum value in 8 days. All qRT-PCR and Western blotting experiments were replicated at least three times. The data represent mean ± SEM (*n* = 3). ** p* < 0.05 and *** p* < 0.01 indicate statistically significant differences compared with the basal level using the one-way ANOVA followed by Bonferroni’s multiple comparison test.

**Figure 2 ijms-24-03251-f002:**
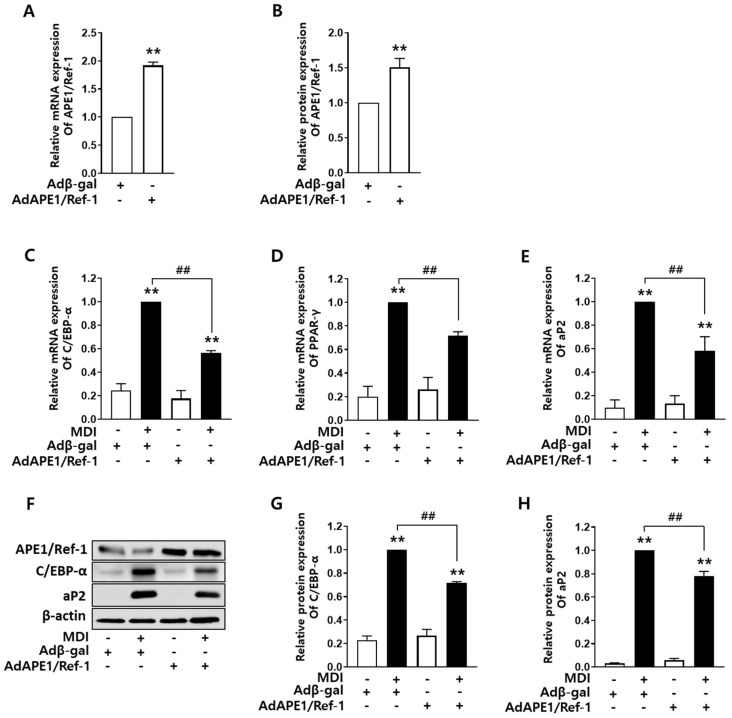
Overexpression of *APE1/Ref-1* reduced adipocyte differentiation of 3T3-L1 cells. (**A**,**B**) 3T3-L1 cells infected with adenovirus expressing *APE1/Ref-1* (AdAPE1/Ref-1) showed increased mRNA and protein levels of *APE1/Ref-1*. (**C**–**E**) The mRNA levels of *C/EBP-α*, *PPAR-γ*, and *aP2* were decreased in *APE1/Ref-1*-overexpresed cells during differentiation. (**F**) Representative Western blotting results for APE1/Ref-1, C/EBP-α, and aP2 protein levels during the differentiation of *APE1/Ref-1*-overexpressed adipocytes. (**G**,**H**) *C/EBP-α* and *aP2* were decreased in *APE1/Ref-1*-overexpressed adipocytes during differentiation. The graph shows the average band intensity normalized to β-actin expression. The relative mRNA and protein levels of *C/EBP-α*, *PPAR-γ*, and *aP2* were calculated compared with those in cells cultured in MDI alone. The data represent mean ± SEM (*n* = 3); *** p <* 0.01 vs. basal level; *^##^ p* < 0.01 vs. MDI alone denotes statistically significant differences using the one-way ANOVA followed by Bonferroni’s multiple comparison test.

**Figure 3 ijms-24-03251-f003:**
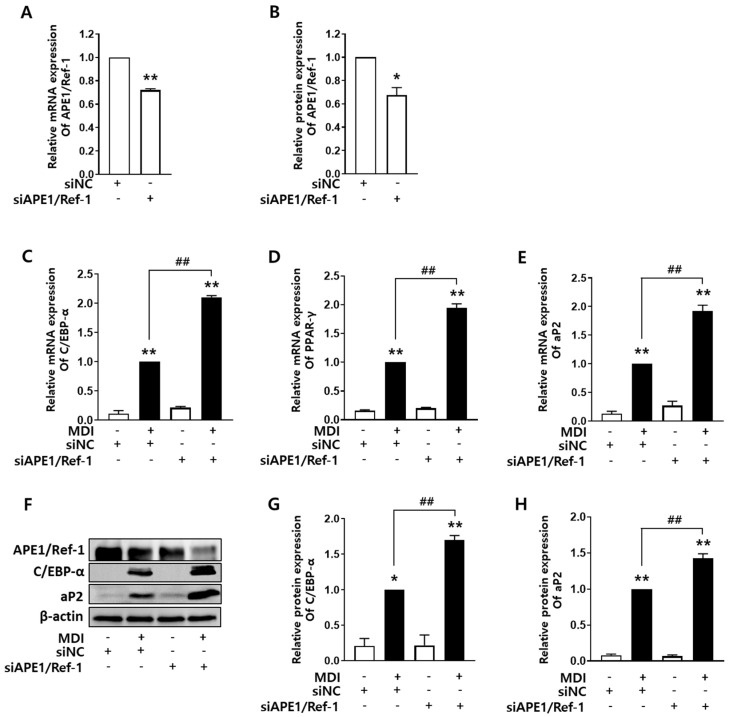
Silencing *APE1/Ref-1* using siRNA (siAPE1/Ref-1) increased adipocyte differentiation of 3T3-L1 cells. (**A**,**B**) *APE1/Ref-1* silencing in 3T3-L1 cells decreased the mRNA and protein levels of *APE1/Ref-1*. (**C**–**E**) In cells transfected with *APE1/Ref-1* siRNA, the mRNA levels of *C/EBP-α*, *PPAR-γ*, and *aP2* increased during adipocyte differentiation. (**F**) Representative Western blotting results of aP2, C/EBP-α, and APE1/Ref-1 protein. (**G**,**H**) C/EBP-α and aP2 protein expression increased in cells transfected with *siAPE1/Ref-1*. The graph shows the average band intensity normalized to β-actin expression. The relative mRNA and protein levels of *C/EBP-α, PPAR-γ,* and *aP2* were calculated compared with those in cells cultured in MDI alone. The results are represented as mean ± SEM; ** p* < 0.05, *** p* < 0.01 vs. basal level; *^##^ p* < 0.01 vs. MDI alone denotes statistically significant differences using the one-way ANOVA followed by Bonferroni’s multiple comparison test.

**Figure 4 ijms-24-03251-f004:**
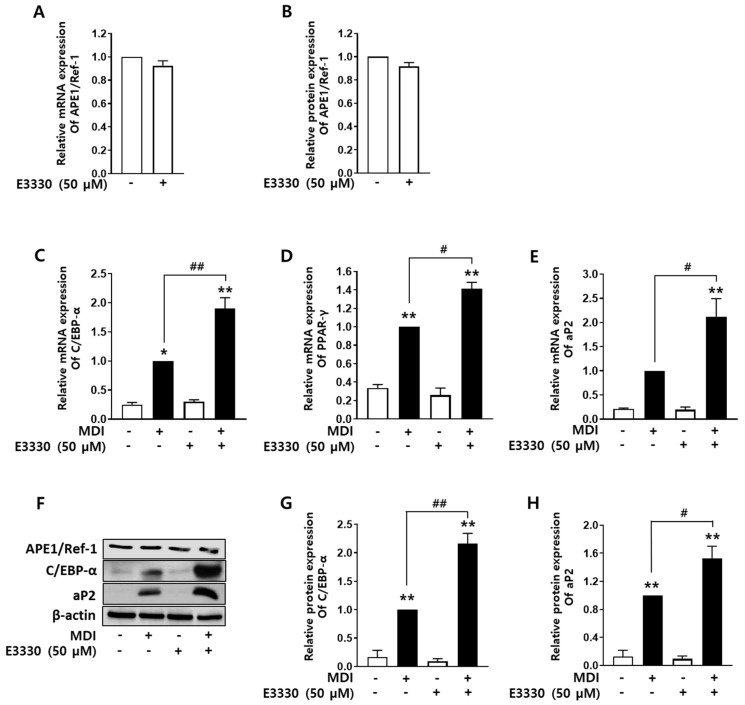
Redox inhibition of *APE1/Ref-1* increased adipocyte differentiation of 3T3-L1 cells. To inhibit the redox activity of *APE1/Ref-1*, the cells were incubated with 50 µM E3330 for 24 h before differentiation induction using the MDI cocktail. (**A**,**B**) E3330 did not affect the mRNA and protein levels of *APE1/Ref-1*. (**C**–**E**) The mRNA levels of *C/EBP-α*, *PPAR-γ*, and *aP2* were increased with the redox inhibition of *APE1/Ref-1* using E3330 in differentiating adipocytes. (**F**) Representative Western blot results of aP2, C/EBP-α, and APE1/Ref-1 protein levels upon redox inhibition of APE1/Ref-1 using E3330 in differentiating adipocytes. (**G**,**H**) Protein levels of C/EBP-α and aP2 increased upon redox inhibition of APE1/Ref-1 expression using E3330 in differentiating adipocytes. The graph shows the average band intensity normalized to β-actin level. The relative mRNA and protein levels of *C/EBP-α*, *PPAR-γ*, and *aP2* were calculated compared with those in cells cultured in MDI alone. The results are represented as mean ± SEM; ** p* < 0.05, *** p* < 0.01 vs. basal level; *^#^ p* < 0.05, *^##^ p* < 0.01 vs. MDI alone denotes statistically significant differences using the one-way ANOVA followed by Bonferroni’s multiple comparison test.

## Data Availability

Not applicable.

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
