# Peer review of "APE1/Ref-1 Inhibits Adipogenic Transcription Factors during Adipocyte Differentiation in 3T3-L1 Cells"

_ijms, 2023, doi:10.3390/ijms24043251_

Round 1

Reviewer 1 Report

This is a study demonstrating the inverse relationship between APE/Ref1 and adipogenic transcription factors during differentiation of 3T3-L1 cells. APE1/Ref-1 over-expression inhibited adipocyte differentiation by suppressing C/EBP-α, and PPAR-γ, and aP2 in the cells. The methods used are sound and conclusions are supported by the results. The presentation is clear and readily understood. There are just a few minor edits to awkward sentences and minor typographical errors (for example, lines 17, 86, 239, 301).

Author Response

Responses to the comments of Reviewer 1

• Throughout Methods section, state replicate numbers for the experiments.
Response: We thank you for the comment. The replicate number for the experiments has been mentioned in the Material and Method.

• Line 17 – “The redox activity of APE1/Ref-1 involved in inflammatory response and regulate transcription factors related to cell survival pathways.” Consider - The redox activity of APE1/Ref-1 is involved in inflammatory responses and regulation of DNA binding of transcription factors related to cell survival pathways.
Response: We thank you for the valuable comment. Per your suggestion, we have revised the sentence as follows: “The redox activity of APE1/Ref-1 is involved in inflammatory responses and regulation of DNA binding of transcription factors related to cell survival pathways” (lines 17).

• Line 86 – “small-interfering RNA of APE1/Ref-1 (siAPE1/Ref-1)….” Should be – Small-interfering RNA of APE1/Ref-1 (siAPE1/Ref-1)…. 
Response: We thank you for detail comment, We have made the necessary revision in the manuscript (line 299).

• Line 211 – “APE1/Ref-1 cells….” Replace with 3T3-L1 cells.
Response: We apologize for the oversight. We have made the necessary revision in the manuscript (line 340).

• Line 239 – “As established that overexpression of APE1/Ref-1 suppressed adipocyte differentiation, the effect of gene silencing of APE1/Ref-1 during MDI-induced adipocyte differentiation were evaluated in 3T3-L1 cells with transfection of APE1/Ref-1 siRNA.” Consider – Having established that overexpression of APE1/Ref-1 suppressed adipocyte differentiation, the effect of gene silencing of APE1/Ref-1 during MDI-induced adipocyte differentiation was evaluated in 3T3-L1 cells with transfection of APE1/Ref-1 siRNA.
Response: We thank you for the comment. We have revised the sentence (line 160) accordingly.

• Line 301- “…transcription factors. And inhibiting the redox function…..” Consider - Conversely, gene silencing of APE1/Ref-1 increased adipocyte differentiation by increasing the expression of adipogenic transcription factors and inhibiting the redox function of….
Response: We thank you for the comment. We have revised the sentence (lines 224–226) accordingly.

• Line 313 – “There is some evidence that APE1/Ref-1 inhibits adipocyte differentiation by acting as a transcription cofactor.” That statement should have a citation or consider removing the word “adipocyte”.
Response: We thank you for the comment. We have cited appropriate references for the sentence: “There is some evidence that APE1/Ref-1 inhibits differentiation by acting as a transcription cofactor [Lee, K.M. (2017), Chen, T. (2015), Domenis, R. (2014)].” Further, the term “adipocyte” has been deleted in lines 239–240.

Reviewer 2 Report

Dear author, The manuscript is well written. FIrstly,Obesity is a common concern at present. Secondly, The author not only studied the expression rule of APE1/Ref-1 in the process of adipocyte differentiation, but also studied the effect of APE1/Ref-1 gene on adipocyte differentiation by over-expression and gene silencing. Finally, the effect of APE1/Ref-1 inhibitor on adipocyte differentiation was studied,and APE1/Ref-1 inhibited adipocyte differentiation by reduce the relative mRNA and protein of transcription factors such as CEBPα, PPARγ and AP2.But there are a few issues that need to be clarified before publication:

Point 1: line62-71: In the preface, it is suggested to increase the research progress of APE1/Ref-1 related to adipose tissue and adipocytes.

Point 2: line79: What kind of cell does it mean? Please seriously consider whether it is “mouse adipose tissue-derived 3t3-L1 cells”? Are Mouse 3T3-L1 fibroblasts?

Point 3: P166: Check carefully whether the "p" in p<0.05 and p<0.01 is italicized in the whole text.

Point 4: Fig1-G:It is suggested to supplement the relative protein expression of CEBPα during adipocyte differentiation.

Point 5: line211:It is suggested to replace “APE1/Ref-1 cells” with 3T3-L1 cells.

Point 6: Figure 1-C, D, E and Figure 2-A, B have no error lines. Are they three biological repetitions?

Point 7:The discussion content is relatively simple. It is suggested to increase the discussion to explain how APE1/Ref-1 mediates transcription factors to participate in adipogenesis.

Point 8: Keywords are not reasonable, please Modify and replace others.

Round 2

Reviewer 2 Report

Hello, author, all my questions have been solved, and there are no other questions.